# The Association of Influencer Marketing and Consumption of Non-Alcoholic Beer with the Purchase and Consumption of Alcohol by Adolescents

**DOI:** 10.3390/bs13050374

**Published:** 2023-05-03

**Authors:** Chun-Yin Hou, Tzu-Fu Huang, Fong-Ching Chang, Tsu-En Yu, Tai-Yu Chen, Chiung-Hui Chiu, Ping-Hung Chen, Jeng-Tung Chiang, Nae-Fang Miao, Hung-Yi Chuang

**Affiliations:** 1Department of Health Promotion and Health Education, National Taiwan Normal University, Taipei 10610, Taiwan; punny63@gmail.com (C.-Y.H.);; 2Graduate Institute of Information and Computer Education, National Taiwan Normal University, Taipei 10610, Taiwan; 3The Graduate Institute of Mass Communication, National Taiwan Normal University, Taipei 10610, Taiwan; 4Department of Statistics, National Chengchi University, Taipei 11605, Taiwan; 5Post-Baccalaureate Program in Nursing, Taipei Medical University, Taipei 11031, Taiwan; 6Department of Public Health, Kaohsiung Medical University, Kaohsiung 80708, Taiwan

**Keywords:** adolescent, non-alcoholic beer, alcohol, influencer marketing, parental mediation

## Abstract

In this study, we examined influencer marketing and consumption of non-alcoholic beer by adolescents to determine how these factors could affect the intentions of adolescents to purchase and drink alcohol. A total of 3121 high-school students recruited from 36 schools in Taiwan completed a self-administered questionnaire during the COVID-19 pandemic in 2022. The results indicate that 19% of these adolescents consumed non-alcoholic beer and 28% consumed alcohol in the past year. Multivariate analysis positively associated adolescents’ exposure to influencer marketing with their purchase and consumption of non-alcoholic beer. Adolescents’ exposure to influencer marketing of non-alcoholic beer combined with lower levels of parental restrictive mediation was associated with increased odds of the purchase and consumption of alcohol. For individuals who did not purchase alcohol in the past year, both the exposure to influencer marketing and the consumption of non-alcoholic beer were associated with intending to purchase alcohol in the future. Similarly, individuals who previously abstained from the consumption of alcohol, both the exposure to influencer marketing and the consumption of non-alcoholic beer were associated with intending to consume alcohol. In conclusion, when adolescents were exposed to influencer marketing of non-alcoholic beer they were more likely to consume it, which resulted in an increased likelihood that they would then purchase and consume alcohol.

## 1. Introduction

Underage drinking is an emerging public health problem. Globally, alcohol use was the seventh leading risk factor for both deaths and disability-adjusted life years (DALYs) in 2016. Among the population aged 15–49 years, 12.2% of male deaths and 3.8% of female deaths are attributed to the consumption of alcohol [1]. Both early initiation and heavy consumption of alcohol throughout adolescence are associated with increased risks of alcohol-related harm [2]. Abuse of alcohol by adolescents is associated with poor health outcomes across the lifespan [3]. A review study associated adolescents’ binge drinking with poorer cognitive functioning in areas such as learning, memory, executive functioning, and impulsivity [4]. A longitudinal study has also associated even low levels of alcohol consumption with the risk for later abuse of cannabis [5].

The availability and sales of non-alcoholic beer have increased in several countries [6,7]. There is debate as to whether the consumption of non-alcoholic beer serves as a substitute for alcohol or as a gateway to its consumption. Some studies show that increasing the availability of non-alcoholic beers seems to substitute for the purchase of alcoholic products [8,9]. Other studies indicate that alcohol-free liquor products are being marketed to increase consumption among children, adolescents, and pregnant women rather than being marketed to lower the consumption of traditional alcoholic products. These studies also stress the intent to increase the acceptable number of occasions that alcohol-free liquor products might be considered for consumption (i.e., weekday lunchtimes and sports/fitness activities) [10,11]. In Japan, about 30% of advertisements for alcohol-flavored non-alcoholic beverages were aired during times that typically restrict alcohol advertising [12]. Since most non-alcoholic beer products are made to appear very similar to alcoholic beer packaging, experts worry about the risks involved when children and adolescents consume drinks that look, smell, and taste like alcohol. Some scholars have raised concerns that the marketing of zero-alcohol beverages might serve as a gateway to the consumption of alcohol among minors and have suggested conducting more research to aid in establishing meaningful policy regulations [13].

As children and adolescents spend more time online and using social media [14], they are increasingly exposed to the digital marketing of non-alcoholic products that mimic alcohol. Studies have indicated that exposure to such products on social media is related to increased intention to consume alcohol [15,16]. Review studies have shown that adolescents and young adults who view digital alcohol marketing by clicking on alcohol ads, visiting alcohol-branded websites, and liking or sharing ads on social media were more likely to consume alcohol [17] and to later develop alcohol-related problems [18]. A study of four countries also showed that greater exposure to online alcohol marketing was related to drinking initiation and binge drinking [19]. In other studies, content analysis of Instagram posts has shown that the majority of influencers who post about alcohol show positive and appealing characteristics, such as positive emotional experiences, achievement, individuality, and camaraderie [20,21]. Compared with brand posts, influencer posts lead to more brand “likes” [22]. Future research is needed to examine the impact of influencer marketing on alcohol consumption among minors [20].

Parents have an important role to play in reducing adolescents’ risk for initiating the consumption of alcohol. Studies have positively associated social media alcohol content exposure with the consumption of alcohol and shown that parental mediation reduces this relationship [23,24]. A review study positively associated the risk of adolescent consumption of alcohol with parental provision of alcohol, favorable parental attitudes towards alcohol consumption, and parental drinking. The same study negatively associated consumption of alcohol with parental monitoring, high-quality parent-child relationships, parental support, and parental involvement [25]. A study found that youth whose parents critiqued media messages reported more critical thinking skills, less interaction with alcohol brands on social media, and less interest in alcohol [26]. Parenting factors and peer influences in early adolescence should be considered valuable exercises to reduce the risk of later alcohol-related harm [2].

In Taiwan, children and adolescents are spending more time using social media for entertainment and communication. A national survey in Taiwan showed that about 57% of children and adolescents use Instagram and spend 1.5 h per day on the platform [27]. The advertising of non-alcoholic beer and alcohol products in the form of social media influencer marketing has increased in Taiwan. However, there are no clear regulations that require influencers disclose sponsored content, and many young influencers endorse non-alcoholic beer and associated products on social media. Although Taiwanese law bans the sale or delivery of alcohol to children and adolescents, the Taiwan Youth Health Survey has shown that half of middle school students and about three-fourths of high school students have consumed alcohol [28]. School teachers complained that students bring non-alcoholic beer products to schools to share with classmates. A study conducted in Taiwan has associated adolescent exposure to alcohol advertising in the media with both the initiation and the persistence of alcohol consumption [29]. Experts are concerned that children and adolescents are more susceptible to influencer marketing and non-alcoholic beer products that look identical to their alcoholic counterparts.

Due to COVID-19, children and adolescents have increased their time online and tend to be more vulnerable to mental distress [30,31]; alcohol companies are known to market their products and brands on social media platforms related to COVID-19 [32]. The World Health Organization (WHO) has stated that the alcohol industry has used increasingly sophisticated advertising and promotion techniques, such as linking alcohol brands to sports, sponsorships and social media, to gain new customers (i.e., youth) [33]. Despite studies showing the effect that social media and influencer marketing has on the intake of unhealthy foods and beverages among minors [34,35,36,37], relatively few studies have been conducted in Asian societies to examine the relationships between exposure to the influencer marketing of non-alcoholic beer and the impact of non-alcoholic beer use on the intentions of adolescents to consume alcohol. Thus, the present study examines the relationship of exposure to the influencer marketing and consumption of non-alcoholic beer with adolescents’ intention to purchase and consume alcohol during the COVID-19 pandemic.

## 2. Materials and Methods

### 2.1. Participants

Data for this cross-sectional study were collected from 36 schools in urban and rural areas in Taiwan during the COVID-19 pandemic in 2022. A probability proportionate to size sampling method was used to systematically draw a random sample of schools. Four to six classes in each of the sample schools were selected. The sampling frame was a list of schools and high-school student enrollments from the Ministry of Education 2021 academic year data. The participating schools in urban areas included two schools in Taipei City, four schools in New Taipei City, and eleven schools in Kaohsiung City, while the participating schools in rural areas included eleven schools in Pingtung County, six schools in Yilan County, and two schools in Hualien County. Due to the COVID-19 pandemic, outside persons were not allowed to enter the schools. Thus, we contacted the teachers who were in charge of students’ health issues in each of the sample schools, explained the survey, and asked whether they could help. We asked the teachers to explain the survey and provide the students with consent forms and questionnaires so they could participate their classrooms. Overall, a total of 3251 high school students (aged 15–18) completed the self-administered questionnaires. After excluding some questionnaires with missing data, 3121 were deemed valid (boys: 1472; girls: 1649; urban: 1356; rural: 1765). The response rate for this survey was about 89%.

### 2.2. Materials

The self-administered questionnaire was developed based on previous studies. A group of six experts were invited to assess the content validity of the questionnaire. The questionnaires were delivered to students in their classrooms. A pretest survey was conducted to interpret the students’ responses to the survey and to evaluate the reliability of the scales in the questionnaire. Students were assured that their information would be protected and anonymous. Approval was obtained from the Institutional Review Board at National Taiwan Normal University (202012HS010).

### 2.3. Measures

#### 2.3.1. Non-Alcoholic Beer/Alcohol/Cigarette Use

Non-alcoholic beer/alcohol/tobacco use was measured with three items. Students were asked “How often do you drink non-alcoholic beer?”, “How often do you drink alcohol?”, and “How often do you smoke cigarettes?” Response options for each item included “never”, “ever (ever before a year)”, “seldom (a few times within a year)”, “sometimes (a few times within a month)”, and “usually (a few times within a week)”. If students answered “a few times within a year” or more frequently, they were coded as a non-alcoholic beer/alcohol drinker or cigarette user.

#### 2.3.2. Non-Alcoholic Beer/Alcohol Purchase

Non-alcoholic beer/alcohol purchase was measured with two items. Students were asked “How often do you purchase non-alcoholic beer?”, and “How often do you purchase alcohol?” Response options for each item included “never”, “ever (ever before a year)”, “seldom (a few times within a year)”, “sometimes (a few times within a month)”, and “usually (a few times within a week)”. If students answered “a few times within a year” or more frequently, they were coded as a non-alcoholic beer/alcohol buyer.

#### 2.3.3. Non-Alcoholic Beer/Alcohol Drinking Intentions

Non-alcoholic beer/alcohol drinking intentions were measured using two items. Students’ drinking intentions were measured based on their answers regarding to whether they thought they would drink non-alcoholic beer or alcohol in the next year. Response options for each item included “definitely not”, “probably not”, “probably yes”, and “definitely yes”. If students answered, “probably yes” or “definitely yes”, they were coded as intending to drink non-alcoholic beer/alcohol.

#### 2.3.4. Non-Alcoholic Beer/Alcohol Purchase Intentions

Intention to purchase and consume non-alcoholic beer/alcohol was measured using two items. Students’ purchase intentions were measured based on their answers to whether they thought they would buy non-alcoholic beer or alcohol in the next year. Response options for each item included “definitely not”, “probably not”, “probably yes”, and “definitely yes”. If students answered, “probably yes” or “definitely yes”, they were coded as intending to purchase non-alcoholic beer/alcohol.

#### 2.3.5. Influencer Marketing Exposure

Influencer marketing exposure was measured using two items. Students were asked “During the past year, how often did you see influencers promote non-alcoholic beer?” and “During the past year, how often did you see influencers promote alcohol products?” Response options for each item were rated on a 5-point scale: “never” (scoring 1), “seldom” (scoring 2), “sometimes” (scoring 3), “usual” (scoring 4), and “always” (scoring 5). Higher scores indicated greater levels of influencer marketing exposure.

#### 2.3.6. Parental Mediation

Parental mediation measures were adapted from a previous study [38] and consisted of 24 items (Cronbach’s α = 0.95). Parental mediation included three types: parental restrictive mediation (8 items, Cronbach’s α = 0.87), parental active mediation (11 items, Cronbach’s α = 0.96), and parental monitoring mediation (5 items, Cronbach’s α = 0.88). Response options for each item included “never” (scoring 1), “seldom” (scoring 2), “sometimes” (scoring 3), “usually” (scoring 4), and “always” (scoring 5). Higher scores indicated greater levels of parental mediation.

#### 2.3.7. Demographic Variables

Demographic information includes gender (male vs. female), area (rural vs. urban), grade (10th grade, 11th grade, 12th grade), and household income (lower-income/lower-middle income class vs. median/upper income class).

### 2.4. Statistical Analysis

Statistical analyses were performed using SAS version 9.4 for Windows. Percentages and means were calculated for all variables. Both *t*-tests and Chi-squared tests was conducted to analyze gender differences in non-alcoholic beer/alcohol consumption and purchase, influencer marketing exposure, and parental mediation. A series of generalized estimating equation (GEE) models were conducted to examine the factors related to non-alcoholic beer and alcohol consumption and purchases, and to intention to drink and purchase non-alcoholic beer and alcohol.

## 3. Results

### 3.1. Non-Alcoholic Beer and Alcohol Purchase and Drinking Behavior

Overall, 57% of participant adolescents were from rural areas, while 20% lived in household poverty. About 15% of adolescents (boys: 16%, girls: 14%) had purchased non-alcoholic beer in the past year and 19% (boys: 18%, girls: 19%) had drunk non-alcoholic beer in the past year. In addition, 19% of adolescents (boys: 21%, girls: 18%) had purchased alcohol in the past year and 28% (boys: 29%, girls: 28%) had drunk alcohol in the past year. By gender, the rate for boys who had purchased alcohol in the past year was significantly higher than that for girls (Table 1).

### 3.2. Exposure to Influencer Marketing of Non-Alcoholic Beer and Alcohol

Based on the results analysis, the mean score of exposure to influencer marketing of non-alcoholic beer was 2.06 (boys = 2.06, girls = 2.06), with 32% of adolescents reporting sometimes, often, or always being exposed to influencer marketing of non-alcoholic beer. The mean score of exposure to influencer marketing of alcohol was 2.28 (boys = 2.34, girls = 2.22), with 41% of adolescents reporting sometimes, often, or always being exposed to influencer marketing of alcohol. By gender, boys reported greater levels of exposure to influencer marketing of alcohol (Table 2).

### 3.3. Non-Alcoholic Beer/Alcohol Purchase and Drinking Intentions

Overall, 31% of adolescents reported intending to buy non-alcoholic beer in the next year, while 34% of adolescents reported intending to drink non-alcoholic beer. In addition, 32% of adolescents reported intending to purchase alcohol, while 40% of adolescents reported intending to drink alcohol. By gender, more girls reported intending to drink non-alcoholic beer and alcohol (Table 1).

### 3.4. Factors Related to Non-Alcoholic Beer/Alcohol Purchase and Drinking

For all adolescents, GEE results indicated that adolescents who lived in an urban area, were at the 12th grade level, had smoked, and had more exposure to influencer marketing of non-alcoholic beer were more likely to purchase non-alcoholic beer. Similarly, female students who lived in an urban area, were at the 12th grade level, had smoked, and had more exposure to influencer marketing of non-alcoholic beer were more likely to drink non-alcoholic beer (Table 3).

For all adolescents, GEE results indicated that adolescents who were at the 12th grade level, had smoked, had more exposure to influencer marketing of alcohol, drank non-alcoholic beer, and had less parental restrictive mediation were more likely to purchase and to drink alcohol (Table 4).

### 3.5. Factors Related to Non-Alcoholic Beer/Alcohol Purchase and Drinking Intentions

For adolescents who did not purchase non-alcoholic beer, GEE results indicated that females at the 12th grade level who had more exposure to influencer marketing of non-alcoholic beer were more likely to intend to purchase non-alcoholic beer. For adolescents who did not consume non-alcoholic beer, GEE results indicated that female students at the 12th grade level who had more exposure to influencer marketing of non-alcoholic beer were more likely to intend to drink non-alcoholic beer (Table 5).

For adolescents who did not purchase alcohol, GEE results indicated that adolescents who lived in an urban area, were at the 11th and 12th grade levels, had smoked, had more exposure to influencer marketing of alcohol, drank non-alcoholic beer, and had less parental restrictive mediation were more likely to intend to purchase alcoholic drinks. For adolescents who did not drink alcohol, GEE results indicated that females who lived in an urban area, were at the 12th grade level, had more exposure to influencer marketing of alcohol, and drank non-alcoholic beer were more likely to intend to drink alcohol (Table 6).

## 4. Discussion

The results of this study show that more than one-fourth of adolescents had consumed alcohol in the past year. Adolescents’ exposure to influencer marketing of alcohol was associated with the purchase and drinking of alcohol. These results were consistent with a previous study that positively associated exposure to social influencers’ alcohol-related posts (alcohol posts) with the drinking of alcohol by adolescents [20]. Prior studies have also associated adolescents’ exposure to alcohol-related social media and digital alcohol marketing with alcohol consumption [15,16,17,19]. In response to these findings, researchers have suggested regulating digital marketing [36,37], and the WHO has taken the position that alcohol marketing presents cross-border challenges in the form of influencer marketing [33]. In 2018 Lithuania enacted a law on alcohol control to provide a comprehensive ban on alcohol advertising, which included all digital media, but court rulings have been contradictory regarding the potential utilization of zero-alcohol beverages by alcohol companies to promote the sale of alcoholic beverages [39]. Future research is needed to examine whether the marketing of non-alcoholic beer increases alcohol consumption by minors.

In addition, in the present study we found that about one-fifth of adolescents had consumed non-alcoholic beer in the past year, which was associated with the consumption of alcohol when influencer marketing of alcohol was considered. Both the consumption of non-alcoholic beer and exposure to the influencer marketing of alcohol have also been positively associated with adolescents’ intentions to consume alcohol. Prior studies have established these behavioral patterns in both children and adolescents [40,41]. These findings support the notion that alcohol-free liquor products seem to be marketed to children and adolescents [10,11]. Since most non-alcoholic beer products look identical to their alcoholic counterparts, children could have difficulty differentiating between the two. Experts are concerned about the risk for children and adolescents in consuming a drink that looks, smells, and tastes like alcohol. It is logical to assume that developing an early habit of consuming non-alcoholic drinks that have the other characteristics of alcoholic drinks is likely to transition to regularly drinking alcohol later. The Czech Republic launched a campaign called “Don’t Hop Children” to warn parents of the risks associated with giving children non- or low-alcoholic beers.

The results of the present study show that adolescent exposure to influencer marketing of non-alcoholic beer and alcoholic drinks is common and is positively associated with non-alcoholic beer/alcohol consumption and purchase. Another study has shown that influencers have a significant impact on social comparison, materialism, and the fear of being excluded, which affects the intention to purchase of endorsed products [42]. Children and adolescents may not be aware that the sponsored influencer posts are actually advertising, which makes them more vulnerable to persuasive attempts [43,44]. At least one study has shown that only a few alcohol posts disclose advertisement information, and that posts with sponsorship disclosures yield fewer likes and comments than posts without such disclosures [20]. Several studies have suggested mandating disclosures of influencer marketing to increase children’s and adolescents’ advertising literacy and reduce advertising effects [44,45,46,47]. Some countries, such as the United States [48] and the United Kingdom [49], have published guidelines requiring that social media influencers disclose material connections they have with brands they endorse.

Moreover, the results of the present study negatively associated parental restrictive mediation with adolescents’ alcohol consumption and purchase. Prior studies have positively associated social media exposure with alcohol consumption among adolescents, while showing that parental mediation moderates the association between social media exposure and consumption of alcohol [23,24]. At least one study has shown that the combination of written and spoken sponsorship disclosure information and an active parental mediation style increases adolescents’ cognitive advertising literacy [50]. Pediatricians could guide parents and children to develop digital literacy skills to prevent or mitigate negative outcomes [51]. Many studies suggest implementing media literacy enhancement programs and parental mediation to strengthen critical thinking skills and prevent online risks [26,52].

Furthermore, results of the present study show that boys purchase alcohol at higher rates than girls, while girls have higher rates of intention to drink non-alcoholic beer and alcoholic drinks. A separate study involved analysis of posts concerning alcohol on social media and found that drinking was presented as a feminine practice, which could appeal to a wider range of women [53]. Another study showed that lower-strength alcoholic products tend to target non-traditional consumers, such as adolescents and pregnant women, in an effort to extend the range of consumers [10]. Future studies could further analyze influencer marketing regarding non-alcoholic beer and alcohol content and assess the advertising effects on boys and girls, respectively. At least one study has shown that strengthening social media literacy has a protective role for adolescents, and for girls in particular, and that it reduces the damaging effects of exposure to idealized images on social media [54]. Schools could develop and implement gender-specific programs and involve parents in social media literacy educational interventions to prevent the consumption of alcohol and reduce negative impacts from social media marketing.

The present study had some limitations. First, this was a cross-sectional study and could not demonstrate causality for topics such as influencer marketing exposure, non-alcoholic beer drinking, and alcohol consumption among adolescents. Future studies could conduct longitudinal studies to examine the causal relationship between the consumption of non-alcoholic beer and the drinking of alcohol. Second, self-reporting of influencer marketing exposure and non-alcoholic beer/alcohol drinking are prone to potential recall bias. Third, 12th grade students had a higher rate of refusing to participate in the survey compared with 10th and 11th grade students due to college entrance exam preparation. Selection bias must be considered. Fourth, the survey was conducted over six months, and the nature of change during adolescence may have created bias in the interpretation of the results. Finally, social desirability bias could have influenced the truthfulness of adolescents’ reports of alcohol consumption, meaning that the rates of adolescents’ drinking could be underestimated. However, confidentiality was emphasized in this study. Despite these limitations, the strength of the present study was its large sample size which was used to examine the relationships between influencer marketing exposure, parental mediation, and the consumption of non-alcoholic beer, and to establish how these relationships affect adolescents’ intention to purchase and consume alcohol.

## 5. Conclusions

The present study associated exposure to influencer marketing with increased odds of the purchase and consumption of both non-alcoholic beer and alcohol. Drinking non-alcoholic beer was associated with increased odds of drinking alcohol. In addition, for adolescents who did not consume non-alcoholic beer, exposure to influencer marketing was associated with increased odds of intending to drink and purchase non-alcoholic beer. In a similar manner, for adolescents who did not drink alcohol, exposure to influencer marketing and non-alcoholic drinking was associated with increased odds of intending to drink and purchase alcohol. In addition, parental restrictive mediation was negatively associated with adolescent intention to purchase alcohol, as well as with the intention to consume alcohol. Future studies could examine potential causal relationships between non-alcoholic beer and the drinking of alcohol. Governments should regulate social media influencer marketing to prevent negative health consequences, while schools could develop social media literacy programs to strengthen adolescents’ coping skills to resist influencer marketing on social media that promotes the consumption of alcohol. Finally, community programs could be implemented to enhance parental mediation.

## Figures and Tables

**Table 1 behavsci-13-00374-t001:** Non-alcoholic beer and alcohol purchase and consumption by gender.

Variable	Overall		Girls		Boys		*p* Value
	n	%	n	%	n	%	
Area							0.742
Urban	1356	43.45	721	43.72	635	43.14	
Rural	1765	56.55	928	56.28	837	56.86	
Grade							<0.0001 ***
10th	1282	41.08	691	41.91	591	40.15	
11th	1088	34.86	626	37.96	462	31.39	
12th	751	24.06	332	20.13	419	28.46	
Household poverty							<0.0001 ***
Yes	642	20.57	282	17.10	360	24.46	
No	2479	79.43	1367	82.90	1112	75.54	
Cigarette smoking							<0.0001 ***
Yes	139	4.47	47	2.86	92	6.28	
No	2970	95.53	1597	97.14	1373	93.72	
Non-alcoholic beer purchase							0.212
Yes	466	14.98	234	14.22	232	15.83	
No	2645	85.02	1411	85.78	1234	84.17	
Non-alcoholic beer drinking							0.822
Yes	576	18.51	307	18.66	269	18.35	
No	2535	81.49	1338	81.34	1197	81.65	
Alcohol purchase							0.007 **
Yes	603	19.42	289	17.62	314	21.43	
No	2502	80.58	1351	82.38	1151	78.57	
Alcohol drinking							0.746
Yes	880	28.32	461	28.08	419	28.60	
No	2227	71.68	1181	71.92	1046	71.40	
Intention to purchase non-alcoholic beer							0.0003 ***
Yes	951	30.58	549	33.39	402	27.42	
No	2159	69.42	1095	66.61	1064	72.58	
Intention to drink non-alcoholic beer							<0.0001 ***
Yes	1068	34.39	626	38.12	442	30.19	
No	2038	65.61	1016	61.88	1022	69.81	
Intention to purchase alcohol							0.4838
Yes	993	31.94	516	31.39	477	32.56	
No	2116	68.06	1128	68.61	988	67.44	
Intention to drink alcohol							0.0047 **
Yes	1247	40.11	698	42.46	549	37.47	
No	1862	59.89	946	57.54	916	62.53	

Note: Chi-square tests conducted. Overall n = 3121; girl n = 1649; boy = 1472. ** *p* < 0.01, *** *p* < 0.001.

**Table 2 behavsci-13-00374-t002:** Influencer marketing exposure, and parental mediation by gender.

Variable	Overall		Girls		Boys		*p* Value
	Mean	SD	Mean	SD	Mean	SD	
Exposure to influencer marketing of non-alcoholic beer	2.06	1.06	2.06	1.03	2.06	1.09	0.906
Exposure to influencer marketing of alcohol	2.28	1.09	2.22	1.04	2.34	1.14	0.005 **
Parental restrictive mediation	1.87	0.88	1.82	0.87	1.92	0.89	0.002 **
Parental active mediation	1.16	0.59	1.20	0.59	1.12	0.58	0.0003 ***
Parental monitoring mediation	1.43	0.69	1.40	0.64	1.47	0.75	0.002 **

Note: *t*-tests conducted. Overall n = 3121; girl; n = 1649; boys = 1472. ** *p* < 0.01, *** *p* < 0.001.

**Table 3 behavsci-13-00374-t003:** Factors related to non-alcoholic beer purchase and consumption.

Variable	Non-Alcoholic Beer Purchase		Non-Alcoholic Beer Drinking	
	OR	95% CI	OR	95% CI
Gender (male vs. female)	0.95	(0.77–1.17)	0.84	(0.72–0.99)
Area (rural vs. urban)	0.79	(0.65–0.96)	0.79	(0.68–0.91)
Grade (grade 11 vs. grade 10)	1.20	(0.97–1.50)	1.17	(0.95–1.43)
Grade (grade 12 vs. grade 10)	1.52	(1.17–1.97)	1.40	(1.15–1.72)
Household poverty (yes vs. no)	1.00	(0.78–1.28)	0.91	(0.69–1.20)
Cigarette smoking	7.66	(4.93–11.9)	7.15	(4.99–10.25)
Exposure to influencer marketing of non-alcoholic beer	1.67	(1.55–1.80)	1.58	(1.47–1.69)
Parental restrictive mediation	1.01	(0.87–1.17)	0.99	(0.84–1.16)
Parental active mediation	1.11	(0.92–1.34)	1.11	(0.91–1.36)
Parental monitoring mediation	1.09	(0.91–1.30)	1.06	(0.88–1.27)

Note: Using GENMOD program with REPEATED statement, binomial distribution, link = logit, exchangeable. Cluster: school n = 36, observation n = 3059 (Non-alcoholic beer purchase model), 3059 (Non-alcoholic beer drinking model).

**Table 4 behavsci-13-00374-t004:** Factors related to alcohol purchase and consumption.

Variable	Alcohol Purchase		Alcohol Drinking	
	OR	95% CI	OR	95% CI
Gender (male vs. female)	1.17	(0.90–1.51)	0.94	(0.76–1.16)
Area (rural vs. urban)	0.94	(0.68–1.29)	0.83	(0.64–1.09)
Grade (grade 11 vs. grade 10)	1.17	(0.85–1.62)	1.18	(0.92–1.51)
Grade (grade 12 vs. grade 10)	1.72	(1.06–2.80)	1.46	(1.03–2.06)
Household poverty (yes vs. no)	1.12	(0.87–1.43)	0.97	(0.78–1.21)
Cigarette smoking	15.13	(9.07–25.22)	10.88	(6.3–18.81)
Exposure to influencer marketing of alcohol	1.29	(1.20–1.39)	1.26	(1.19–1.34)
Non-alcoholic beer drinking	8.85	(6.85–11.43)	11.87	(9.58–14.70)
Parental restrictive mediation	0.82	(0.69–0.98)	0.80	(0.70–0.93)
Parental active mediation	0.90	(0.69–1.18)	1.06	(0.84–1.33)
Parental monitoring mediation	1.05	(0.85–1.30)	1.00	(0.82–1.22)

Note: Using GENMOD program with REPEATED statement, binomial distribution, link = logit, exchangeable. Cluster: school n = 36, observation n = 3052 (Alcohol purchase model), 3054 (Alcohol drinking model).

**Table 5 behavsci-13-00374-t005:** Factors related to intention to drink and purchase non-alcoholic beer.

Variable	Non-Alcoholic Beer Purchase Intention		Non-Alcoholic Beer Drinking Intention	
	OR	95% CI	OR	95% CI
Gender (male vs. female)	0.70	(0.60–0.82)	0.70	(0.59–0.83)
Area (rural vs. urban)	0.93	(0.74–1.17)	1.01	(0.82–1.25)
Grade (grade 11 vs. grade 10)	1.26	(0.96–1.64)	1.08	(0.83–1.41)
Grade (grade 12 vs. grade 10)	1.81	(1.28–2.55)	1.48	(1.10–1.99)
Household poverty (yes vs. no)	0.85	(0.68–1.07)	0.81	(0.62–1.06)
Cigarette smoking	2.11	(0.98–4.53)	1.97	(0.99–3.94)
Exposure to influencer marketing of non-alcoholic beer	1.23	(1.13–1.34)	1.19	(1.08–1.31)
Parental restrictive mediation	0.99	(0.81–1.21)	0.94	(0.79–1.14)
Parental active mediation	1.00	(0.81–1.23)	1.10	(0.89–1.36)
Parental monitoring mediation	1.07	(0.92–1.24)	0.93	(0.80–1.08)

Note: Using GENMOD program with REPEATED statement, binomial distribution, link = logit, exchangeable. Cluster: school n = 36, observation n = 2601 (Non-alcoholic beer purchase intention model), 2491 (Non-alcoholic beer drinking intention model).

**Table 6 behavsci-13-00374-t006:** Factors related to intention to drink and purchase alcohol.

Variable	Alcohol Purchase Intention		Alcohol Drinking Intention	
	OR	95% CI	OR	95% CI
Gender (male vs. female)	0.96	(0.74–1.26)	0.75	(0.58–0.97)
Area (rural vs. urban)	0.67	(0.48–0.93)	0.72	(0.53–0.97)
Grade (grade 11 vs. grade 10)	1.63	(1.21–2.21)	1.01	(0.74–1.39)
Grade (grade 12 vs. grade 10)	2.53	(1.58–4.04)	1.87	(1.25–2.78)
Household poverty (yes vs. no)	1.00	(0.71–1.42)	0.79	(0.60–1.04)
Cigarette smoking	4.52	(1.85–11.08)	2.64	(0.96–7.24)
Exposure to influencer marketing of alcohol	1.22	(1.11–1.35)	1.20	(1.07–1.35)
Non-alcoholic beer drinking	2.85	(2.01–4.05)	2.15	(1.52–3.06)
Parental restrictive mediation	0.82	(0.68–0.98)	0.88	(0.74–1.05)
Parental active mediation	1.13	(0.88–1.46)	1.03	(0.81–1.31)
Parental monitoring mediation	0.99	(0.81–1.21)	0.90	(0.74–1.1)

Note: Using GENMOD program with REPEATED statement, binomial distribution, link = logit, exchangeable. Cluster: school n = 36, observation n = 2455 (Alcohol purchase intention model), 2188 (Alcohol drinking intention model).

## Data Availability

The data presented in this study can be requested and provided.

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
