# Peer review of "The Association of Influencer Marketing and Consumption of Non-Alcoholic Beer with the Purchase and Consumption of Alcohol by Adolescents"

_behavsci, 2023, doi:10.3390/bs13050374_

Round 1

Reviewer 1 Report

Title: Impact of Exposure to Influencer Marketing and Non-alcoholic Beer on Alcohol Use and Intention to Drink among Adolescents

This cross-sectional study estimated the associations between exposure to influencer marketing and non-alcoholic bear on underage alcohol use and susceptibility among adolescents from 36 high schools in Taiwan. Overall, this study is very well written. I have some comments for the authors to consider or justify.  

Title: “Impact” is way too causal, particularly given that the study findings are based on a cross-sectional study. I would recommend the authors to use “The associations” instead, and also revise through the manuscript to avoid causal language.

Introduction is good. I would recommend the authors to include more background information regarding the Taiwan regulations and rates, if any, on underage alcoholic use, non-alcoholic beer use, social media use, advertising on social media, sponsored advertising, etc. among adolescents, for audience who are not familiar with this topic in Taiwan.

Methods:

More information on PPS is needed. After schools were selected, are there any other stages of sampling (e.g., grades, classes, etc.) within each school?

Since PPS is used to generate a probability sample, I wonder has sample weights been calculated and how?

Students’ intention to purchase and drink alcohol were measured in a 4-point Likert scale and treated as a continuous variable in regression models. I wonder if the authors had conducted residual analysis, because this may be problematic given the potential violation of linearity and normality assumptions. I would recommend the authors to conduct ordinal regression and check the proportional odds assumption or further categorize this variable.

I also recommend the authors to conduct multilevel analysis to account for the potential clustered correlations, given that students are nested within schools, and schools nested within districts.

Did the questionnaire include SES factors and other underage substance use (i.e., like tobacco products, drugs, etc.), that needs to be controlled as potential covariates?

Results:

Table 1: I wonder why there is a smaller proportion of 12th-grade students compared with the other two grades. Is it due to differential attrition rates? If so, may acknowledge it in limitation.

Some languages are causal. For example, instead of using “increase”, the authors may consider using “was associated with increased odds”.

Discussion looks good.

Author Response

Dear Editor:

We appreciate the reviewers’ helpful comments. We have made changes in response to the comments. We are grateful for your help.

Responses to Reviewer 1

This cross-sectional study estimated the associations between exposure to influencer marketing and non-alcoholic bear on underage alcohol use and susceptibility among adolescents from 36 high schools in Taiwan. Overall, this study is very well written. I have some comments for the authors to consider or justify.  

  1. Title: “Impact” is way too causal, particularly given that the study findings are based on a cross-sectional study. I would recommend the authors to use “The associations” instead, and also revise through the manuscript to avoid causal language.

Thank you for your compliment and suggestion. We have changed the title into “The associations of influencer marketing and consumption of non-alcoholic beer with the purchase and consumption of alcohol by adolescents”. In addition, we also revised the manuscript to avoid causal language.

  1. Introduction is good. I would recommend the authors to include more background information regarding the Taiwan regulations and rates, if any, on underage alcoholic use, non-alcoholic beer use, social media use, advertising on social media, sponsored advertising, etc. among adolescents, for audience who are not familiar with this topic in Taiwan.

Thank you for this suggestion. We have added more background information regarding the rates of alcohol drinking among adolescents and alcohol regulation in Taiwan.

  1. Methods:
    More information on PPS is needed. After schools were selected, are there any other stages of sampling (e.g., grades, classes, etc.) within each school?
    Since PPS is used to generate a probability sample, I wonder has sample weights been calculated and how?

Thank you for this suggestion. We have added more information on PPS in the Methods section. We did not use the sampling weights. We added selection bias in the Limitations section.

  1. Students’ intention to purchase and drink alcohol were measured in a 4-point Likert scale and treated as a continuous variable in regression models. I wonder if the authors had conducted residual analysis, because this may be problematic given the potential violation of linearity and normality assumptions. I would recommend the authors to conduct ordinal regression and check the proportional odds assumption or further categorize this variable.

Thank you for this suggestion. We have changed the intention variables into a binary level to examine the odds in the results section (Tables 5, 6).

  1. I also recommend the authors to conduct multilevel analysis to account for the potential clustered correlations, given that students are nested within schools, and schools nested within districts.

Thank you for this suggestion. We have changed all the multivariate analyses to Generalized Estimating Equation (GEE) with clusters of schools within cities/counties in the Results section.

  1. Did the questionnaire include SES factors and other underage substance use (i.e., like tobacco products, drugs, etc.), that needs to be controlled as potential covariates?

Thank you for this suggestion. We have added household income and cigarette smoking variables in the Generalized Estimating Equation (GEE) models in the Results section.

  1. Results:
    Table 1: I wonder why there is a smaller proportion of 12th-grade students compared with the other two grades. Is it due to differential attrition rates? If so, may acknowledge it in limitation.

Thank you for this suggestion. We have added a description regarding a possible reason for the smaller proportion of 12th grade students and potential selection bias in the limitations section.

  1. Some languages are causal. For example, instead of using “increase”, the authors may consider using “was associated with increased odds”.

Thank you for this suggestion. We have revised the manuscript.

  1. Discussion looks good.

Thank you for your comments.

We are grateful for your help.

Reviewer 2 Report

Line 97- random sampling was used for the selection of school. Sample frame will be helpful to clarify this issue.

I suggest the age range of the sample  be mentioned.

It is not clear  what inform the total number of 3121 sample size for the current study.

A brief socio-economic status  background of the population under study could be helpful to other readers given the fact that the study was carried out both in the rural and urban areas.

In line 158  replace serious with series.

Administration of the questionnaire during data collection process in the classroom could be helpful

Line 136-141- The duration of a year could introduce bias in the interpretation of the results given the growing nature of adolescence. I suggest it be included in the limitation of the study

Author Response

Dear Editor:

We appreciate the reviewers’ helpful comments. We have made changes in response to the comments. We are grateful for your help.

Responses to Reviewer 2

  1. Line 97- random sampling was used for the selection of school. Sample frame will be helpful to clarify this issue.
    I suggest the age range of the sample  be mentioned.

Thank you for this suggestion. We have added a description regarding the sample frame in the Methods section. We also added the age range of the sample in the Methods section.

  1. It is not clear what inform the total number of 3121 sample size for the current study.

Thank you for this suggestion. We have added some descriptions regarding the sample size in the Methods section.

  1. A brief socio-economic status background of the population under study could be helpful to other readers given the fact that the study was carried out both in the rural and urban areas.

Thank you for this suggestion. We have added a description regarding socio-economic status and urban/rural background in the Results section.

  1. In line 158  replace serious with series.

Thank you for this suggestion. We corrected the error.

  1. Administration of the questionnaire during data collection process in the classroom could be helpful

Thank you for this suggestion. We have added a description regarding the data collection process in the classrooms in the Methods section.

  1. Line 136-141- The duration of a year could introduce bias in the interpretation of the results given the growing nature of adolescence. I suggest it be included in the limitation of the study

Thank you for this suggestion. We have discussed bias in the limitations section.

We are grateful for your help and kindness.

Round 2

Reviewer 1 Report

I appreciate the authors' efforts to address my concerns. I have no further comments.